# Pd(OAc)$_2$-catalyzed asymmetric hydrogenation of sterically hindered N-tosylimines

Jianzhong Chen[1], Zhenfeng Zhang[1], Bowen Li[1], Feilong Li[2], Yulin Wang[2], Min Zhao[2], Ilya D. Gridnev [3], Tsuneo Imamoto[4] & Wanbin Zhang [1]

Asymmetric hydrogenation of sterically hindered substrates still constitutes a long-standing challenge in the area of asymmetric catalysis. Herein, an efficient palladium acetate (an inexpensive Pd salt with low toxicity) catalyzed asymmetric hydrogenation of sterically hindered N-tosylimines is realized with high catalytic activities (S/C up to 5000) and excellent enantioselectivities (ee up to 99.9%). Quantum chemical calculations suggest that uniformly high enantioselectivities are observed due to the structurally different S- and R-reaction pathways.

[1] Shanghai Key Laboratory for Molecular Engineering of Chiral Drugs, School of Chemistry and Chemical Engineering, Shanghai Jiao Tong University, 800 Dongchuan Road, Shanghai 200240, China. [2] School of Chemistry and Molecular Engineering, East China University of Science and Technology, Shanghai 200237, China. [3] Department of Chemistry, Graduate School of Science, Tohoku University, Aramaki 3-6, Aoba-ku, Sendai 9808578, Japan. [4] Department of Chemistry, Graduate School of Science, Chiba University, Chiba 263-8522, Japan. Correspondence and requests for materials should be addressed to W.Z. (email: wanbin@sjtu.edu.cn)

Sterically hindered chiral amines have found widespread application for the asymmetric syntheses of bioactive substances, drugs, and ligands (Fig. 1a)[1–15]. Over the past half-century, significant progress has been made in the catalytic asymmetric hydrogenation of various imines for the synthesis of chiral amines[16–23]. Imines bearing relatively small substituents, such as methyl or ethyl groups directly connected to the carbon atom of a C=N group, have been widely used as substrates in asymmetric hydrogenation for the preparation of chiral amines, providing excellent stereoselectivities[16–27]. In sharp contrast, the asymmetric hydrogenation of imines bearing bulky substituents (such as the *t*-butyl group) has proved to be far more challenging, even though this methodology provides a straightforward approach for the preparation of chiral bulky amines, important structural elements found in pharmaceuticals, ligands, and other functional molecules. As evidenced by previous studies, the increased bulk of the alkyl substituents results in reduced yields and enantioselectivities of the chiral amine products (Fig. 1b). In 1996, a ruthenium-catalyzed asymmetric hydrogenation of N-tosylimines was described by Charette and Giroux giving the corresponding products in yields of 82%, 80%, <5%, and ees of 62%, 84%, 17%, with methyl, ethyl, and isopropyl substituents, respectively (R' = Ph)[24]. In 2006 and 2007, the groups of Zhang[28] and Zhou[29] developed a palladium trifluoroacetate-catalyzed asymmetric hydrogenation of N-tosylimines. Enantioselectivities decreased from 99% to 93% and from 96% to 88%, respectively, when a methyl group was replaced by an ethyl group (R' = Ph).

Previous methods of choice for the preparation of these chiral bulky amines (only one or two examples) have relied on chiral substrates, auxiliaries, or resolving agents, and suffered from low enantioselectivities and/or yields[30–38]. For example, in 2007, the catalytic enantioselective addition of HN₃ to ketenes was disclosed by Fu and co-workers to give chiral methyl(2,2-dimethyl-1-phenylpropyl)carbamate with 76% ee[33]. In 2009, Zhang and co-workers developed the iridium-catalyzed asymmetric hydrogenation of 2,2-dimethyl-1-phenylpropan-1-imine hydrochloride and 3,3-dimethylbutan-2-imine hydrochloride with 80% ee and 17% ee, respectively[34]. Thus, an efficient methodology for the synthesis of chiral bulky amines by asymmetric hydrogenation remains elusive and highly desired.

Although palladium-catalyzed homogeneous asymmetric hydrogenations of C=C, C=O, and C=N bonds have been extensively investigated, Pd is less commonly used compared to other transition-metals such as Ru, Rh, and Ir, because Pd catalysts are usually less efficient (S/C ratio of no more than 1000)[21,39–43]. In addition, almost all these hydrogenations use Pd(OCOCF₃)₂ as a catalytic precursor, whereas inexpensive Pd salts, such as Pd(OAc)₂, have not exhibited high catalytic activities in the previously reported asymmetric hydrogenation reactions[28,29,44].

Our group have searched for novel approaches for the preparation of important chiral substances using metal-catalyzed asymmetric hydrogenation[45–52]. Recently, we developed a palladium-catalyzed asymmetric hydrogenation and hydrogenolysis of α-acyloxy ketones with excellent yields and high S/C

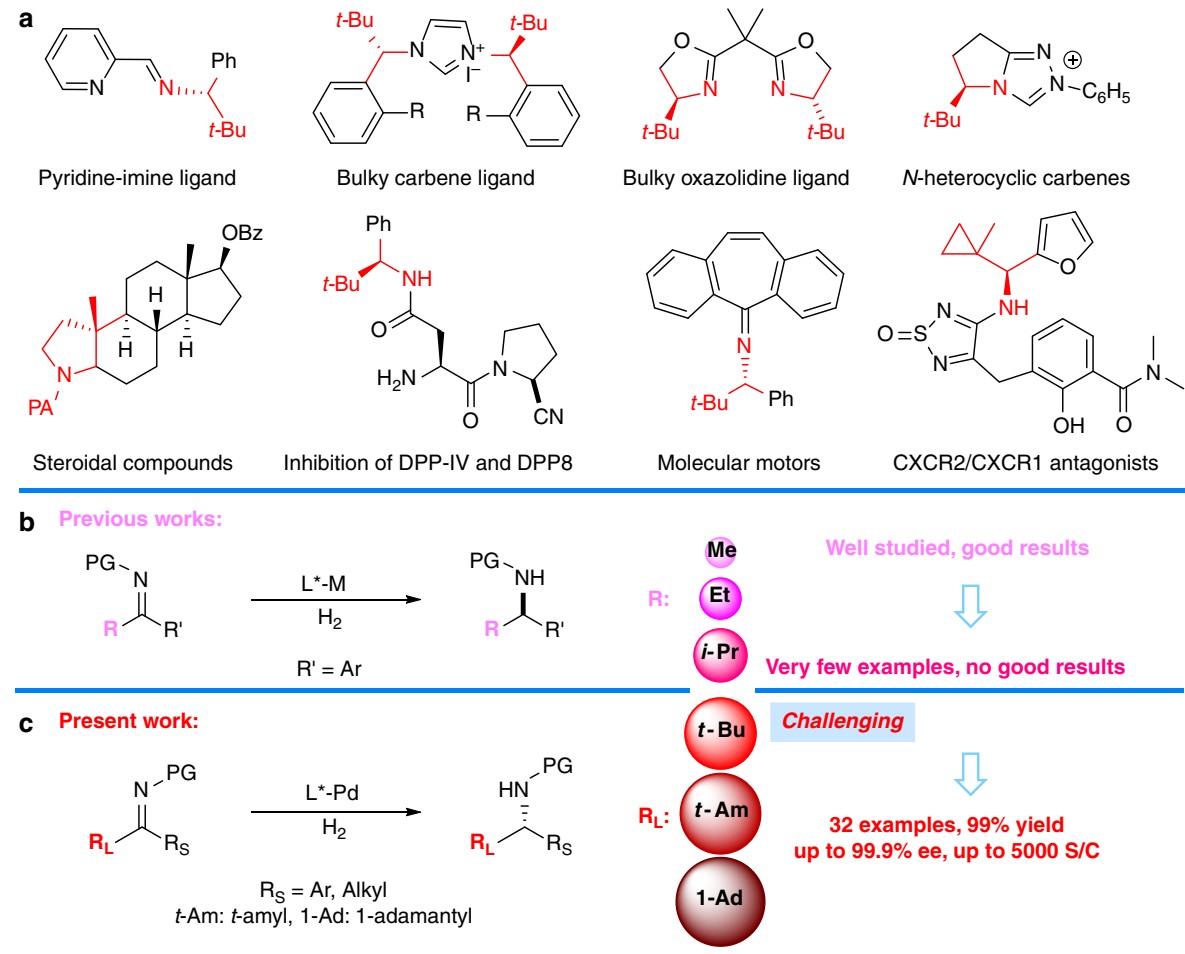

**Fig. 1** Asymmetric hydrogenation of imines for preparation of chiral amines. **a** Representative chiral ligands and bioactive compounds bearing sterically hindered chiral amine skeletons. **b** Previous work about asymmetric hydrogenation of imines. **c** This work: Pd-catalyzed asymmetric hydrogenation of sterically hindered imines

**Table 1 Reaction optimization**

| Entry[a] | Ligand | Pd source | Yield %[b] | ee %[c] |
|---|---|---|---|---|
| 1 | (R)-BINAP | Pd(TFA)$_2$ | 24 | - |
| 2 | (R)-SegPhos | Pd(TFA)$_2$ | <5 | - |
| 3 | (R)-DTBM-SegPhos | Pd(TFA)$_2$ | >99 | 92.2 |
| 4 | (R,R)-QuinoxP* | Pd(TFA)$_2$ | >99 | 99.9 |
| 5 | (R,R)-QuinoxP* | Pd(OAc)$_2$ | >99 | 99.9 |
| 6 | (R,R)-QuinoxP* | PdBr$_2$ | -[d] | - |
| 7 | (R,R)-QuinoxP* | PdCl$_2$ | -[d] | - |
| 8[e] | (R,R)-QuinoxP* | Pd(OAc)$_2$ | >99 | 99.9 |
| 9[f] | (R,R)-QuinoxP* | Pd(OAc)$_2$ | >99 | 99.9 |
| 10[g] | (R,R)-QuinoxP* | Pd(OAc)$_2$ | >99 | 99.9 |

(R)-BINAP

(R)-SegPhos: Ar = Ph
(R)-DTBM-SegPhos:
Ar = 3,5-di-t-Bu-4-MeOC$_6$H$_2$

(R,R)-QuinoxP*

[a]Conditions: **1a** (0.2 mmol), PdX$_2$ (2.0 mol %), ligand (2.1 mol %), TFE (2.0 mL), H$_2$ (40 atm), RT, 24 h, unless otherwise noted
[b]Determined by [1]H NMR analysis
[c]The ee values were determined by HPLC using chiral columns
[d]A mixture of complex by-products was generated
[e]0 °C.
[f]H$_2$ (1 atm)
[g]**1a** (0.70 g), S/C = 1000. Pd(TFA)$_2$ = Pd(OCOCF$_3$)$_2$
The chiral (bis)phosphine we used was abbreviated as (R,R)-QuinoxP* by the inventor. Generally speaking, the significance of '*' is chirality.

(up to 5000–6000) using the bulky ligand DTBM-SegPhos[51,52]. In continuation of the work, we herein report a high yielding and highly enantioselective palladium acetate-catalyzed asymmetric hydrogenation of sterically hindered N-tosylimines (Fig. 1c). A possible reaction mechanism has been proposed via quantum chemical calculations.

## Results

**Investigation of reaction conditions.** Initially, a model asymmetric hydrogenation of (Z)-t-butyl phenyl N-tosylimines (**1a**) was carried out under 40 atm H$_2$ pressure at room temperature in TFE using Pd(TFA)$_2$.

The commonly used ligand, (R)-BINAP, could catalyze this reaction but only gave the desired hydrogenation product with 24% conversion (Table 1, entry 1). The reaction was sluggish when another commonly used ligand, (R)-SegPhos, was used (Table 1, entry 2). However, the bulky ligand (R)-DTBM-SegPhos gave the desired product with almost quantitative conversion and high enantioselectivity (92.2% ee, Table 1, entry 3). To our delight, the hydrogenation proceeded smoothly and the product was obtained with up to 99.9% ee using an electron-rich P-stereogenic diphosphine ligand (R,R)-QuinoxP* (Table 1, entry 4), which has been found to be an efficient chiral diphosphine for Rh- and Ru-catalyzed asymmetric hydrogenations since first being reported in 2005[53–57]. Different Pd precursors were also screened in combination with (R,R)-QuinoxP*. Pd(OAc)$_2$, which is inexpensive, low toxic, and is not commonly used in asymmetric hydrogenations, provided excellent catalytic activity and enantioselectivity (over 99% conversion and 99.9% ee, Table 1, entry 5). Some by-products were produced when using other PdX$_2$-type salts such as PdBr$_2$ and PdCl$_2$ (Table 1, entries 6, 7). The reaction temperature and hydrogenation pressure were also examined. Lowering the temperature to 0 °C had no effect on the reaction conversion and enantioselectivity (Table 1, entry 8). To our surprise, under 1 atm H$_2$ pressure, the reaction proceeded smoothly and gave the product in quantitative conversion and 99.9% ee (Table 1, entry 9). When the S/C ratio was increased to 1000, the product was obtained with no loss in enantioselectivity and full conversion (Table 1, entry 10).

The influence of solvents on this reaction was also examined (Table 2). Several solvents were studied in order to try and avoid the use of TFE. However, TFE was proved to be superior to these solvents. Alcohols such as MeOH and EtOH showed different reactivities (Table 2, entries 1–3). Compared to the excellent results obtained with TFE, MeOH gave the product in excellent enantioselectivity but only 47% yield. Just a trace amount of product was obtained in EtOH. The low polar solvents THF, toluene, and DCM provided low activities (Table 2, entries 4–6).

**Scope of asymmetric catalysis of hindered N-tosylimines.** Substrate scope of the catalytic system was explored using the optimized reaction conditions and with a relatively low catalyst loading (S/C = 200, Table 3). All the tested t-Bu-N-tosylimine substrates were converted to their corresponding products with

**Table 2 Influence of reaction solvent**

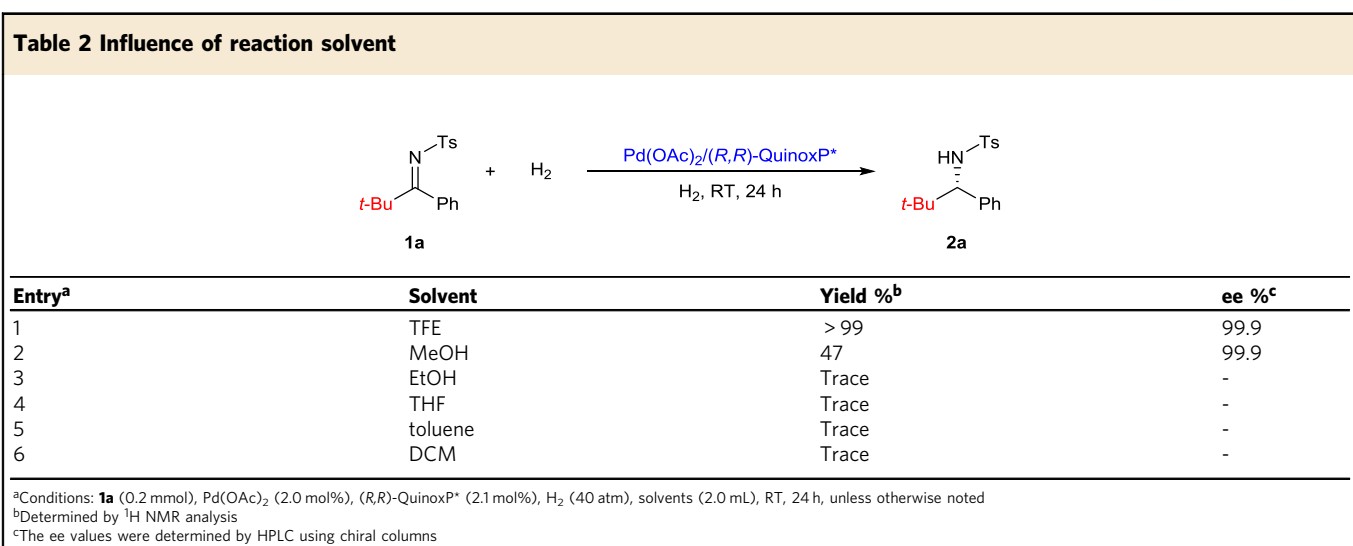

| Entry[a] | Solvent | Yield %[b] | ee %[c] |
|---|---|---|---|
| 1 | TFE | > 99 | 99.9 |
| 2 | MeOH | 47 | 99.9 |
| 3 | EtOH | Trace | - |
| 4 | THF | Trace | - |
| 5 | toluene | Trace | - |
| 6 | DCM | Trace | - |

[a]Conditions: **1a** (0.2 mmol), Pd(OAc)$_2$ (2.0 mol%), (*R*,*R*)-QuinoxP* (2.1 mol%), H$_2$ (40 atm), solvents (2.0 mL), RT, 24 h, unless otherwise noted
[b]Determined by $^1$H NMR analysis
[c]The ee values were determined by HPLC using chiral columns

almost quantitative conversions. The six substrates bearing different R and R$_L$ were hydrogenated in 95–99% yields and with excellent enantioselectivities (**2a**, **b**, **2d–g**, 99.0–99.9% ee), while a substrate bearing a 4-fluoro-substituent gave its corresponding product with 96.0% ee (**2c**). In addition, when the R group was changed to a *t*-Bu group, none of the corresponding product was detected under the standard conditions. Electron-donating and withdrawing substituents at the 3- or 4-position of the benzene ring of R$_S$ did not influence the stereoselectivities (**2h–p**, 99.3–99.9% ee). Similarly, excellent enantioselectivities were obtained for disubstituted substrates, including a substrate bearing a 2-naphthyl group (**2q–u**, 99.4–99.9% ee). It should be noted that even alkyl imine substrates underwent smoothly asymmetric hydrogenation (**2v**, **w**, both 99.8% ee).

**Scope of asymmetric catalysis of functionalized substrates**. To further extend the substrate scope, many functionalized substrates were designed and synthesized. Under standard reaction conditions with a catalyst loading of 0.5 mol %, the products shown in Table 4 were obtained in almost quantitative yields and excellent stereoselectivities. Substrates with different ester groups and different carbon chain lengths were tested under the hydrogenation conditions, giving their corresponding products with excellent results (**2x–z**, 99.7%, 99.4% and 99.1% ee). Changing the ester group to an amide group had no negative impact on the reaction (**2aa**, 97% yield and 99.4% ee). A cyclopentyl-substituted substrate with α-quaternary carbon was also reduced with 99.3% ee (**2ab**). Substrates in which the carbon atom linked to the tetra-substituted carbon center was replaced by an oxygen atom were also reduced with high stereoselectivities, irrespective of whether the oxygen atom was exocyclic or within the ring (**2ac**, **ad**, 99.8% and 99.4% ee). A substrate bearing an ester group at the side of the quaternary carbon could be hydrogenated to the corresponding chiral amine, albeit with slightly lower ee (**2ae**, 96% yield and 96.9% ee). Interestingly a diimine could also be reduced completely with excellent de and ee (**2af**, 98% yield, 99% de and 99.9% ee).

**Synthetic utility of chiral amines products**. The practical utility of this catalytic asymmetric hydrogenation system was evaluated. The catalytic reaction was carried out at a low catalyst loading on a gram scale (**1a**, 3.60 g, $S/C = 5000$, Fig. 2). The reaction proceeded smoothly with quantitative conversion and with 99% ee under 60 atm H$_2$ pressure at 60 °C over 48 h. The chiral compounds **2** have the potential to be used for the synthesis of sterically hindered chiral amines commonly found in optically active substances and chiral ligands[1–15]. Chiral intermediate **4** was obtained in 92% yield and 99% ee by reductive removal of the Ts group of **2a** with sodium naphthalide[58]. This product could be further transformed to the bulky chiral amine structures **5** and **6**[4,9], which can be used as chiral ligands and functional molecular motors. Other ligands, catalysts and bioactive compounds, such as bulky chiral carbene ligands[3], amidoiridium complexes[10], *t*-leucine[31] and CXCR2/CXCR1 antagonists[5], could also be synthesized from compound **4** according to literature procedures.

**Synthetic utility of functionalized chiral amines products**. Furthermore, the ester-functionalized products, **2x** and **2y**, could be converted to chiral γ- and δ-lactams (**8x** and **8y**), which are useful chiral compounds[13–15]. Thus, the compounds of **8** were obtained with high yields and stereochemical fidelity via cyclization to form intermediates (**7x** and **7y**) and subsequent removal of the Ts group, respectively (Fig. 3).

**X-ray crystallographic analysis**. The absolute configurations of the products **2a** and **7x** were determined to be *S* and *R* by X-ray crystallographic analysis (Fig. 4). Therefore, substrates with aryl or alkyl groups (including functionalized compounds) are attacked by the hydride on the same favored side.

**Mechanistic considerations**. Recently the mechanism of the Pd-catalyzed asymmetric hydrogenation of unprotected indoles has been studied in detail[40]. There are two possible reaction pathways, inner-sphere hydrogenation with direct coordination of the C = N bond to Pd, and out-of-sphere hydrogenation with direct involvement of a solvent molecules, both employing a Pd–H complex as a catalyst. In the case of cyclic imines (unprotected indoles) the latter mechanism was computed to be favorable. However, for our sulfonated imines, a similar out-of-sphere mechanism is not possible due to the lack of an NH hydrogen atom that is present in indole substrates. Hence, we computed the catalytic cycle for the inner-sphere hydrogenation of **1a** using Pd–H complex **C1** as a catalyst.

The computed catalytic cycle for the experimentally observed *S*-product exhibited reasonable activation barriers standing well with the experimentally observed reaction rates (Fig. 5a). Thus, a coplanar orientation of the Pd–H and C = N bonds in the adduct **S1** belonging to the *S*-catalytic cycle enables facile hydride

**Table 3 Substrate scope[a]**

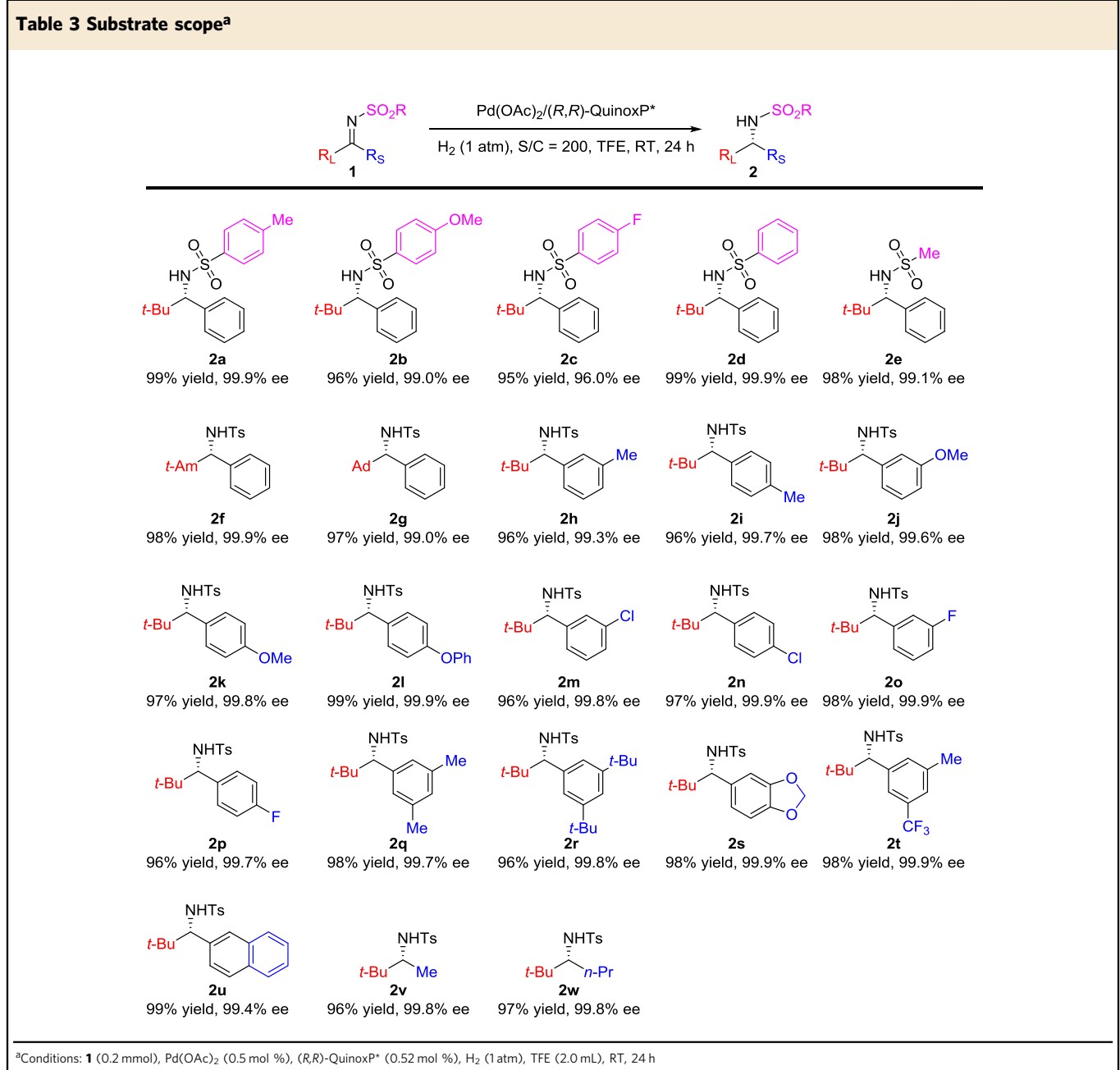

[a]Conditions: **1** (0.2 mmol), Pd(OAc)$_2$ (0.5 mol %), (*R,R*)-QuinoxP* (0.52 mol %), H$_2$ (1 atm), TFE (2.0 mL), RT, 24 h

transfer yielding intermediate **S2**. The latter is further hydrogenated by an additional molecule of H$_2$ providing the product **2a** (*S*) and regenerating the catalyst. Meanwhile, the minor enantiomer *R*-pathways with higher activation barriers was also computed (Fig. 5b).

The asymmetric center is created during the first hydrogenation step and is conserved throughout the catalytic cycle. The second hydrogenation proceeds through **TS2(S),** which is lower in free energy compared with **TS(S)**. Therefore, the level of enantioselection is determined by the relative values of **TS(S)** and **TS(R)**. In Fig. 6, the computed transition states, **TS(S)** and **TS(R)**, for the hydrogen transfer are shown for substrate **1a**. The value of **TS(R)** was computed to be 3.3 kcal mol$^{-1}$ higher in energy than **TS(S)** (Fig. 6, 12.7 vs. 9.4 kcal mol$^{-1}$). Significant differences in stability between the transition states **TS(S)** and **TS(R)** originate from a dissimilarity in the binding modes between the catalyst and the substrate. Thus, the binding of the C = N group coplanar to the Pd–H bond available only in **TS(S)** leads to the formation

of a four-membered ring transition state. Additionally, the numerous weak attractive interactions between the substrate and catalyst are favorable effects in stabilizing the transition state **TS(S)** (see Fig. 6 and Supplementary Table 1)[59–67]. Furthermore, the high catalytic activities of this asymmetric hydrogenation may be partly due to the numerous weak attractive interactions between the substrate and catalyst.

During the optimization of the **TS(R)** structure starting from **TS(S)**, the formation of a six-membered transition state takes place via the sulfonyl group of the substrate due to the fixed geometry of the imine, thus removing the migrating hydride from the plane of the catalyst chelate cycle; this can be seen from the values of the corresponding dihedral angles ((Pd–H–C–N) in Fig. 6b, 74.6° vs. 13.4°). As a result, **TS(S)** is a much "earlier" transition state than **TS(R)** (compare the Pd–H distances in Fig. 6a, 1.56 vs. 1.69 Å). In addition, the substituents of the substrate are further apart from the substituents of the catalyst in the six-membered **TS(R)**, which decreases the stabilizing effect of

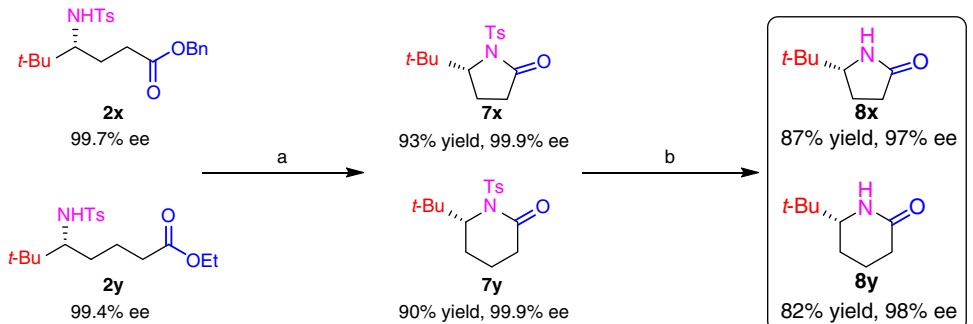

**Table 4 Scope of functionalized substrates[a]**

[Reaction scheme: 1 → 2 with Pd(OAc)₂/(R,R)-QuinoxP*, H₂ (1 atm), S/C = 200, TFE, RT, 24 h]

**2x** 98% yield, 99.7% ee

**2y** 96% yield, 99.4% ee

**2z** 98% yield, 99.1% ee

**2aa** 97% yield, 99.4% ee

**2ab** 96% yield, 99.3% ee

**2ac** 98% yield, 99.8% ee

**2ad** 99% yield, 99.4% ee

**2ae** 96% yield, 96.9% ee

**2af** 98% yield, 99% de, 99.9% ee

[a]Conditions: **1** (0.2 mmol), Pd(OAc)₂ (0.5 mol %), (R,R)-QuinoxP* (0.52 mol %), H₂ (1 atm), TFE (2.0 mL), RT, 24 h

**Fig. 2** Product derivatization. Reagents and conditions are as follows. **a** 1a (3.60 g), $S/C = 5000$, 60 atm H₂, 60 °C, 48 h. **b** 2a (158 mg, 0.5 mmol), naphthalene (5.0 mmol), sodium (5.0 mmol), 1,2-dimethoxyethane (10 mL), −70 °C-RT, 2 h. **c** 2-pyridinecarboxaldehyde, Et₂O, RT, 16 h. **d** 5H-dibenzo[a,d][7]annulen-5-one, TiCl₄, toluene, RT, 24 h

**1a** 3.6 g

**2a** 97% yield, 99% ee

**4** 91% yield, 99% ee

**6** Molecular motors 69% yield, 15:1 de

**5** Pyridine-imine ligand 88% yield, 99% ee

**Fig. 3** Functionalized product derivatization. Reagents and conditions are as follows. **a** 2x, **y** (0.5 mmol), Me₃Al (0.6 mmol), toluene (15 mL), 80 °C, overnight. **b** 7x, **y** (0.5 mmol), naphthalene (5.0 mmol), sodium (5.0 mmol), DME (30 mL), −70 °C, 1 min

**2x** 99.7% ee

**2y** 99.4% ee

**7x** 93% yield, 99.9% ee

**7y** 90% yield, 99.9% ee

**8x** 87% yield, 97% ee

**8y** 82% yield, 98% ee

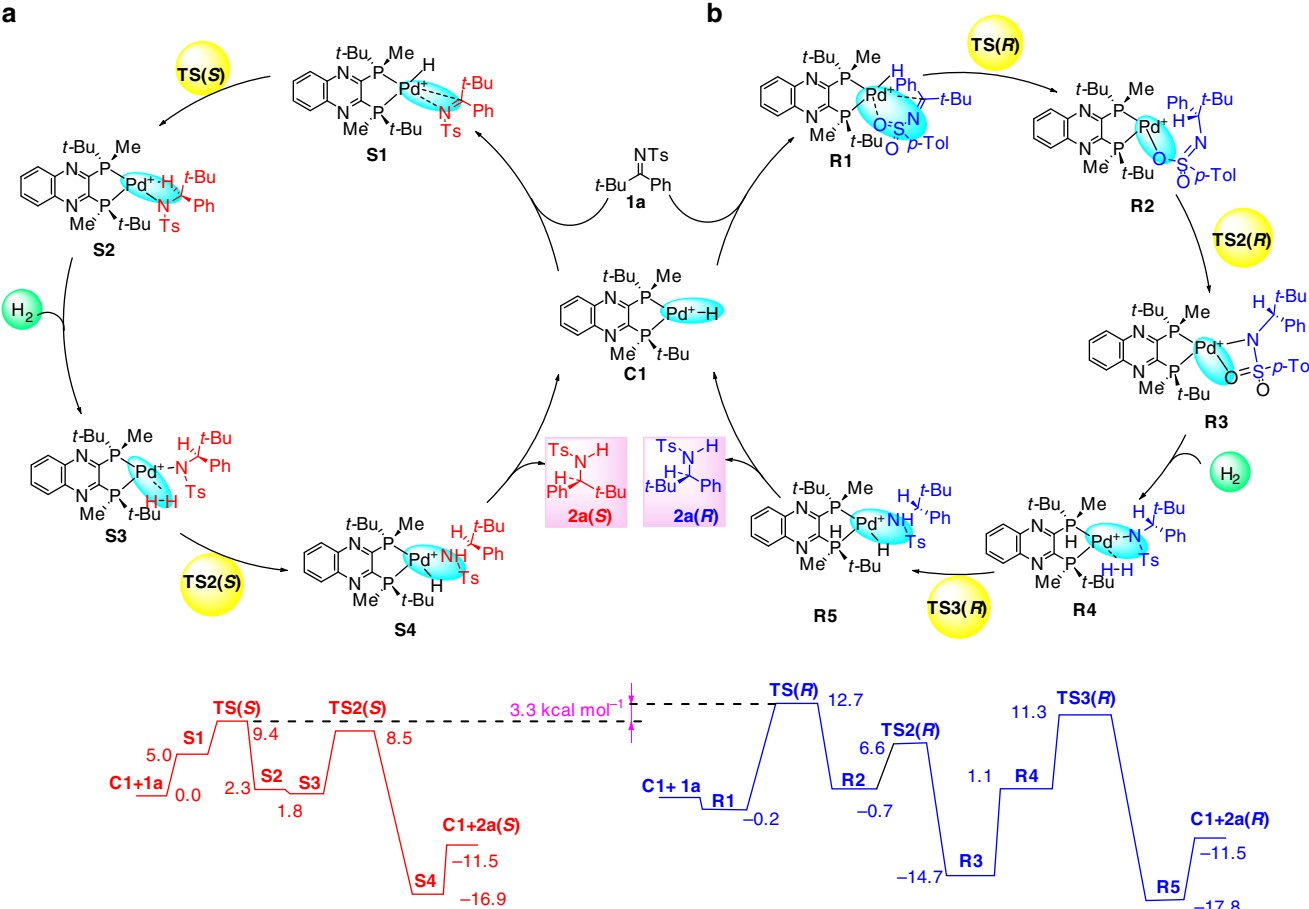

**Fig. 4** X-ray crystallographic analysis. ORTEP representation of **2a** and **7x**

**Fig. 5** Computed catalytic cycles for **1a**. Relative Gibbs free energies in kcal mol$^{-1}$ are shown (WB97XD/6-31g(d,p)/SMD(2,2,2-trifluoroethanol, 298.15 K. 1 atm)

the weak intermolecular interactions (see Fig. 6 and Supplementary Table 1).

## Discussion

In conclusion, a palladium-catalyzed asymmetric hydrogenation of sterically hindered N-tosylimines has been developed. Chiral N-tosylamines were obtained with excellent enantioselectivities (up to 99.9% ee) as well as high yields under 1 atm hydrogen pressure. Palladium acetate, an inexpensive Pd salt with low toxicity, was found to be a suitable catalyst precursor for the homogeneous asymmetric hydrogenation. High catalytic activities were also observed (up to 5000 S/C). The reaction could be conducted on a gram scale and was further applied to the synthesis of useful chiral products and N-ligand. Computations suggested that the enantioselectivity originates from the significant structural differences between the S- and R-pathways. Similarly, excellent enantioselection can be expected for other sulfonated imines possessing a C = N bond with fixed geometry.

## Methods

**Procedure for asymmetric hydrogenation of N-tosylimines**. (R, R)-QuinoxP*
(1.4 mg, 2.1 mol %) and Pd(OAc)$_2$ (0.89 mg, 2.0 mol%) were placed in a dried Schlenk tube under nitrogen atmosphere, and degassed anhydrous acetone (1.0 mL) was added. The mixture was stirred at room temperature for 5 min, then the solvent was removed under vacuum to give the dry catalyst. In a glovebox, substrate **1** (0.2 mmol) was stirred in a solvent (0.5 mL) at room temperature for 10 min. Subsequently, the above-prepared catalyst dissolved in solvent (1.5 mL) was added. The hydrogenation was performed at room temperature under H$_2$ in a stainless steel autoclave for 24 h. After releasing hydrogen, the conversion of the product **2** was determined by $^1$H NMR spectroscopic analysis of the crude reaction mixture. The enantiomeric excesses of the products were determined by HPLC with chiral columns (OD-H, OJ-H, AD-H, or IC-3).

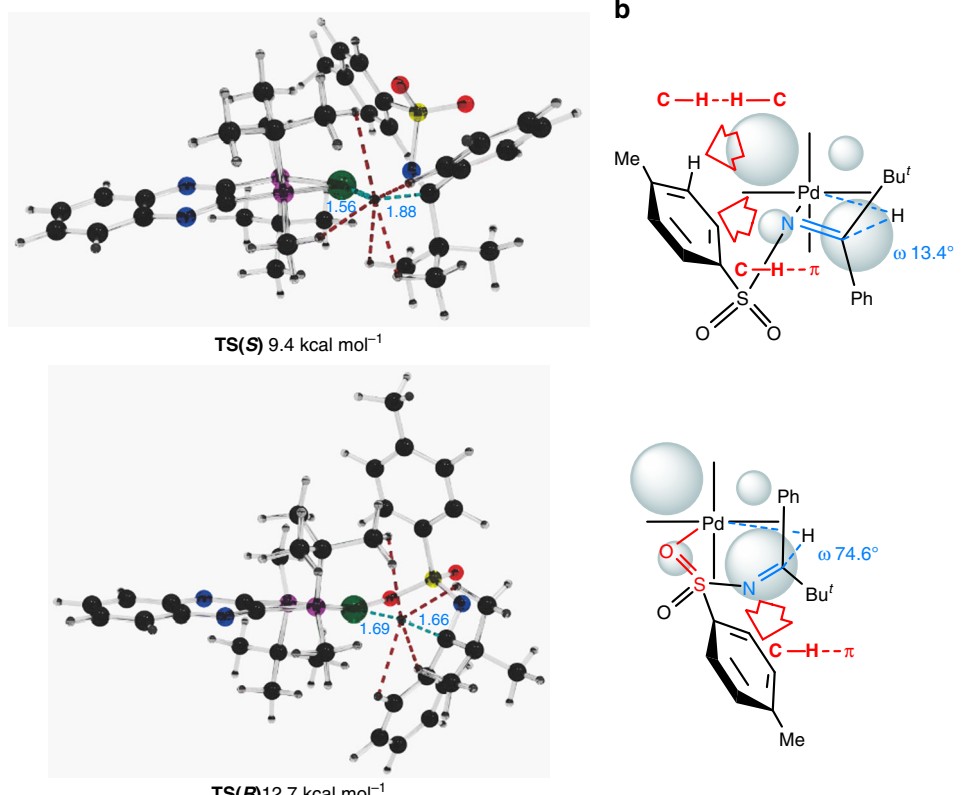

**Fig. 6** The structures of **TS(S)** and **TS(R)**. Computed structures of the transition states for the hydride transfer. (WB97XD/SDD(Pd)/6-31G(d,p)(all others)/SMD(TFE)) (Arrows denote the interactions of two groups)

## Data availability

The authors declare that the data supporting the findings of this study are available within the article and its Supplementary Information file. For the experimental procedures, data of NMR and HPLC analysis and Cartesian coordinates of the optimized structures, see Supplementary Methods and Charts in Supplementary Information file. The X-ray crystallographic coordinates for structures reported in this article have been deposited at the Cambridge Crystallographic Data Center (**2a**: CCDC 1585399, **7x**: CCDC 1585398). These data could be obtained free of charge from The Cambridge Crystallographic Data Center via www.ccdc.cam.ac.uk/data_request/cif.

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

## Acknowledgements

We would like to thank the National Natural Science Foundation of China (Nos. 21620102003 and 21702134), Science and Technology Commission of Shanghai Municipality (Nos. 15JC1402200 and 17ZR1415200), and Shanghai Municipal Education Commission (No. 201701070002E00030) for financial support. We thank the Instrumental Analysis Center of SJTU for characterization. We are grateful to Zhi-Xiang Yu, Peking University, and Yuanyuan Liu, East China Normal University, for helpful discussions concerning our mechanistic studies.

## Author contributions

J.C. conducted most of the synthetic experiments. B.L., F.L., and Y.W. conducted part of the synthetic experiments. I.D.G. conducted the DFT computational study. J.C., Z.Z., I.D.G., and W.Z. wrote the manuscript. Z.Z., M.Z., and T.I. took part in the discussion. W.Z. directed the project.
