## [Peer review file · Nature Communications]

Reviewer #1 (Remarks to the Author):

Although asymmetric hydrogenation of prochiral imines, providing a facile access to biologically active chiral amines, has achieved significant progress, asymmetric hydrogenation of sterically demanding substrates including imines bearing bulky substituents (such as the t-butyl group) remains much less explored. In this manuscript, Zhang and co-workers reported unexpected excellent results on the asymmetric hydrogenation of sterically hindered N-tosylimines catalyzed by chiral palladium-diphosphine complexes. With the easily available palladium acetate and under very mild conditions (1 atm H₂ and room temperature), a wide range of substrates including N-tosylimines and functionalized ones were successfully hydrogenated with high catalytic activities (S/C up to 5000) and excellent enantioselectivities (ee up to 99.9%). This method provides a facile and practical access to sterically hindered chiral amines, which are important chiral building blocks but not easily prepared by other methods. In addition, together with theoretical calculations, the importance of the weak attractive catalyst-substrate interaction in the transition state is demonstrated. This finding is of great importance and represents a breakthrough in the asymmetric hydrogenation of sterically demanding substrates. Thus, these results are valuable to synthetic community and suitable for the publication in this journal after some minor revisions.

1. The Pd-diphosphine catalysts showed remarkable steric effect in this reaction, that is, the sterically demanding diphosphine ligands gave higher activity, up to 5000 TON (an unexpected result for the Pd-catalyst). It is suggested for the authors to give some explanations.
2. Did the authors test other imines bearing N-Boc or N-aryl/alkyl groups?
3. Unlike asymmetric organocatalysis, weak interactions between substrate and catalyst are less possible. Some other recent examples are suggested to be cited (J. Am. Chem. Soc. 2011, 133, 9878; Angew. Chem. Int. Ed. 2012, 51, 5706).
4. Some typos, such as “attractivecatalyst-substrate interactionsin the transition state” to “attractive catalyst-substrate interactions in the transition state”; “Table 4. Scope of alkenylboronic acids” to “Table 4. Scope of functionalized substrates”, etc.

Reviewer #3 (Remarks to the Author):

The authors report a combined experimental and computational study on the Pd-catalyzed asymmetric hydrogenation of sterically hindered N-tosylimines. If TFE is used as a solvent, the reaction allows for the synthesis of the corresponding amines with good yields and enantioselectivities. A significant part of the work is devoted to discuss the origin of the observed enantioselectivity by means of computational investigations. Unfortunately, most of the mechanistic discussion is not convincing and of low technical quality and the conclusions are not supported by

sufficient evidence. Hence, the manuscript does not meet the requirements of Nature Communications.

The discussion regarding the origin of the enantioselectivity relies on the assumption that the reaction proceeds through an inner-sphere mechanism. This is in contrast with previous detailed computational studies on the analogous Pd-catalyzed asymmetric hydrogenation of indoles [J. Am. Chem. Soc. 2014, 136, 7688–7700]. Although the two reactions could in principle proceed through different mechanisms, the authors did not provide sufficient evidence to support their hypothesis.

The authors write: “We have computed both possible pathways using 1a as a substrate. In our case the free activation barrier of the inner-sphere mechanism ($\Delta G^\ddagger = 4.4$ kcal/mol) was computed to be significantly lower than that for the out-of-sphere pathway ($\Delta G^\ddagger = 14.2$ kcal/mol). Hence, we researched for the source of enantioselection within the inner-sphere mechanism.” However, it is not clear how the above activation barriers were computed. Only the hydride migration step that leads to the formation of the stereocenter is discussed in the manuscript and no additional results concerning the following steps are reported in the Supporting Information. The hydride migration step might not be the rate determining step of the reaction for the inner-sphere mechanism. Moreover, the authors did not provide any additional information on the computed out-of-sphere mechanism.

The sampling of the conformational space is an essential part in the modeling of asymmetric transformations. Enantioselectivity in asymmetric catalysis is often driven by a few substrate-catalyst interactions that stabilize selectively one transition state. This aspect is completely neglected in the present work and only one structure for each transition state is reported in the Supporting Information. Note that the TS(S) and TS(R) structures shown in Fig. 5 differ qualitatively. Did the authors try to generate an initial guess for TS(R) starting from the TS(S) structure?

As a final remark, the way the computational details are written in the Supporting Information denotes a clear lack of knowledge of the very basics of computational chemistry. For instance, the authors write “All other atoms were modeled at the 6-31G(d,p) level of theory” or “The solvent effect was accounted for by carrying out optimizations in the SMD force field”.

Reviewer #1

Comment: “Although asymmetric hydrogenation of prochiral imines, providing a facile access to biologically active chiral amines, has achieved significant progress, asymmetric hydrogenation of sterically demanding substrates including imines bearing bulky substituents (such as the t-butyl group) remains much less explored. In this manuscript, Zhang and co-workers reported unexpected excellent results on the asymmetric hydrogenation of sterically hindered *N*-tosylimines catalyzed by chiral palladium-diphosphine complexes. With the easily available palladium acetate and under very mild conditions (1 atm H₂ and room temperature), a wide range of substrates including *N*-tosylimines and functionalized ones were successfully hydrogenated with high catalytic activities (S/C up to 5000) and excellent enantioselectivities (ee up to 99.9%). This method provides a facile and practical access to sterically hindered chiral amines, which are important chiral building blocks but not easily prepared by other methods. In addition, together with theoretical calculations, the importance of the weak attractive catalyst-substrate interaction in the transition state is demonstrated. This finding is of great importance and represents a breakthrough in the asymmetric hydrogenation of sterically demanding substrates. Thus, these results are valuable to synthetic community and suitable for the publication in this journal after some minor revisions.”

Our response:

We appreciate the reviewer's kind comment.

Suggestion 1: The Pd-diphosphine catalysts showed remarkable steric effect in this reaction, that is, the sterically demanding diphosphine ligands gave higher activity, up to 5000 TON (an unexpected result for the Pd-catalyst). It is suggested for the authors to give some explanations. **Our response:**

We appreciate the reviewer's suggestion. As the reviewer has mentioned in the above comment, the reaction activity is very high with sterically demanding diphosphine ligands. We believe that the high catalytic activities of this asymmetric hydrogenation may be partly due to the numerous weak attractive interactions between the substrate and catalyst. We have added a related sentence to the revised manuscript (p11).

Suggestion 2. Did the authors test other imines bearing *N*-Boc or *N*-aryl/alkyl groups?

Our response:

We appreciate the reviewer's suggestion. We have tried to prepare imine substrates bearing *N*-Boc or *N*-aryl/alkyl groups using several methods. For example, we have attempted to prepare imines bearing *N*-Boc or *N*-aryl/alkyl groups from 2,2-dimethyl-1-phenylpropan-1-one with *tert*-butoxycarbonylaminotriphenylphosphine, 4-methoxyaniline, and phenylmethanamine. However, no imine substrates bearing an *N*-Boc group were detected according to NMR analysis. Although a few imine substrates with *N*-aryl/alkyl groups were prepared, they decomposed quickly during the work-up and purification process. These reactions were repeated a number of times; however, no pure imines bearing *N*-Boc or *N*-aryl/alkyl groups were obtained.

Suggestion 3. Unlike asymmetric organocatalysis, weak interactions between substrate and catalyst are less possible. Some other recent examples are suggested to be cited (J. Am. Chem. Soc. 2011, 133, 9878; Angew. Chem. Int. Ed. 2012, 51, 5706).

Our response:

We appreciate the reviewer's suggestion. We have added the two references (ref.62, 63).

Suggestion 4. Some typos, such as “attractivecatalyst-substrate interactionsin the transition state” to “attractive catalyst-substrate interactions in the transition state”; “Table 4. Scope of alkenylboronic acids” to “Table 4. Scope of functionalized substrates”, etc.”

Our response:

We appreciate the reviewer's suggestion. We have corrected them in the revised manuscript (p7).

Reviewer #3

We thank the reviewer for pointing out the problems associated with our computational results - they have allowed us to improve our mechanistic studies. Accordingly, we carried out calculations for the *S*-catalytic cycle. We have discovered that the enantiodetermining step is also “rate-limiting”. This provides important supplementary evidence to our previous mechanistic investigations and compensates for any of the aforementioned deficiencies.

Furthermore, after conducting a number of recalculations, we believe that both the structural different and weak attractive interactions between the substrate and catalyst favor stabilization of the transition states. Thus, in order to ensure our mechanistic conclusions are more accurate and concise, the title of this manuscript has been changed to “*Pd(OAc)₂-Catalyzed Asymmetric Hydrogenation of Sterically Hindered *N*-Tosylimines: Scope and Mechanism*”.

Comment and Suggestion:

“The discussion regarding the origin of the enantioselectivity relies on the assumption that the reaction proceeds through an inner-sphere mechanism. This is in contrast with previous detailed computational studies on the analogous Pd-catalyzed asymmetric hydrogenation of indoles [J. Am. Chem. Soc. 2014, 136, 7688–7700]. Although the two reactions could in principle proceeds through different mechanisms, the authors did not provide sufficient evidence to support their hypothesis.”

Our response:

We appreciate the reviewer's suggestion – this should have been made clear in our initial manuscript. The newly reported reactions cannot in principle proceed through the out-of-sphere

mechanism reported in the above-mentioned paper concerning indoles. Unlike indoles, our substrates do not possess N-H hydrogen atoms that are crucial for the out-of-sphere mechanism computed for indoles. We have made the corresponding adjustments in the text of the paper.

Comment and Suggestion:

“The authors write: “We have computed both possible pathways using 1a as a substrate. In our case the free activation barrier of the inner-sphere mechanism (DG= 4.4 kcal/mol) was computed to be significantly lower than that for the out-of-sphere pathway (DG=14.2 kcal/mol). Hence, we researched for the source of enantioselection within the inner-sphere mechanism.” However, it is not clear how the above activation barriers were computed. Only the hydride migration step that leads to the formation of the stereocenter is discussed in the manuscript and no additional results concerning the following steps are reported in the Supporting Information. The hydride migration step might not be the rate determining step of the reaction for the inner-sphere mechanism. Moreover, the authors did not provide any additional information on the computed out-of-sphere mechanism.”

Our response:

We appreciate the reviewer’s suggestion. We agree that the comparison between inner- and out-of-sphere mechanisms is not entirely accurate and so it has been removed from the discussion. Due to the lack of N-H hydrogen atoms, the reaction cannot in principle proceed via an out-of-sphere mechanism.

In the corrected manuscript, we provide a fully computed catalytic cycle for the *S*-pathway, demonstrating that the first hydride transfer is a rate-determining step, hence the enantioselectivity is determined by the relative stabilities of the transition states **TS(S)** and **TS(R)**, which is discussed in detail in the revised manuscript and Supporting Information.

Comment and Suggestion:

“The sampling of the conformational space is an essential part in the modeling of asymmetric transformations. Enantioselectivity in asymmetric catalysis is often driven by a few substrate-catalyst interactions that stabilize selectively one transition state. This aspect is completely neglected in the present work and only one structure for each transition state is reported in the Supporting Information. Note that the TS(S) and TS(R) structures shown in Fig. 5 differs qualitatively. Did the authors try to generate an initial guess for TS(R) starting from the TS(S) structure?”

Our response:

We all agree to the reviewer’s suggestion that the sampling of the conformational space is an essential part in the modeling of asymmetric transformations, and the mechanical calculations have followed this principle. In our case, in addition to the substrate-catalyst interactions being favorable effects in stabilizing the transition state, the significant structural differences between the **TS(S)** and **TS(R)** play more important roles with regards to enantioselectivity. In addition, enantioselectivities $\geq 99\%$ ee were observed for over 30 structurally different substrates in this reaction. The results may suggest that the origin of the enantioselectivity differs from the most common examples.

As mentioned by the reviewer, **TS(R)** is obtained starting from **TS(S)**. If one simply rotates the sulfonyl group in **TS(S)** by 180 degrees around the C=N bond without touching anything else, a

configuration with a Pd-O distance of 2.45 Å and a Pd-N distance of 2.25 Å is obtained. The nitrogen atom remains the same distance from the Pd and the Pd-O distance is longer. However, if the oxygen atom is positioned ideally to bind with the Pd, the lone pair of the nitrogen atom is directed away from the metal. This configuration of the nitrogen atom is very similar to that seen in **TS(S)**, however, in the latter case, no competition from the oxygen atom of the SO₂ group is observed. Thus, after switching the positions of the Ph and *t*-Bu substituents and subsequent optimization, the Pd-O distance is 2.20 Å and Pd-N distance is 3.38 Å in **TS(R)**. Accordingly, we have added the corresponding description to the last paragraph of the Mechanistic Considerations section.

Comment and Suggestion:

“As a final remark, the way the computational details are written in the Supporting Information denotes a clear lack of knowledge of the very basics of computational chemistry. For instance, the authors write "All other atoms were modeled at the 6-31G(d,p) level of theory" or "The solvent effect was accounted for by carrying out optimizations in the SMD force field".”

Our response:

We thank the reviewer for drawing our attention to the inaccurate terminology. We have changed this passage in the Supporting Information to the following:

Computations were carried out using the range-separated hybrid functional with damped atom-atom dispersion (WB97XD) as implemented in the GAUSSIAN 09 software package. For the palladium atom, the SDD basis set with the associated effective core potential was employed. All other atoms were described with a 6-31G** basis with an additional diffuse function for phosphorus. Non-specific solvation was introduced by using the SMD continuum model (2,2,2-trifluoroethanol).

Yours sincerely,

Prof. Wanbin Zhang
School of Chemistry and Chemical Engineering
Shanghai Jiao Tong University
800 Dongchuan Road, Shanghai 200240, P. R. China
Phone: +86-21-54743265; Fax: +86-21-54743265
E-mail: wanbin@sjtu.edu.cn
Homepage: wanbin.sjtu.edu.cn

Reviewer #3 (Remarks to the Author):

The authors addressed properly all my previous remarks and computed a series of additional calculations. However, I have a few additional comments on the new data presented.

In the new version of the manuscript, they have included in Fig. 5 the computed energy profile leading to the major enantiomer. I would suggest to include in the same figure the energy profile for the minor enantiomer as well (see below). The authors should also include in the caption the energy unit and a summary of the chosen computational methodology and reaction conditions.

Note that TS(S) and TS2(S) are very similar in energy (deviation <1 kcal/mol). This difference is definitely within the error bar of the computational methodology adopted. Hence, the nature of the rate determining step is not clear. This aspect is crucial considering that the first step is reversible according to their calculations. Hence, the second step could (in principle) be the one that determines the enantioselectivity of the reaction. Adding the energy profile for the other enantiomer will clarify this aspect.

Considering the many approximations (necessarily) included in the calculations, I suggest to replace statements like "Computations demonstrated that the enantioselectivity originates from the significant structural differences between the S- and R-pathways." with something like "Computations suggested that the enantioselectivity originates from the significant structural differences between the S- and R-pathways."

Reviewer #3 (Remarks to the Author):

The authors addressed properly all my previous remarks and computed a series of additional calculations. However, I have a few additional comments on the new data presented.

In the new version of the manuscript, they have included in Fig. 5 the computed energy profile leading to the major enantiomer. I would suggest to include in the same figure the energy profile for the minor enantiomer as well (see below). The authors should also include in the caption the energy unit and a summary of the chosen computational methodology and reaction conditions.

Note that TS(S) and TS2(S) are very similar in energy (deviation <1 kcal/mol). This difference is definitely within the error bar of the computational methodology adopted. Hence, the nature of the rate determining step is not clear. This aspect is crucial considering that the first step is reversible according to their calculations. Hence, the second step could (in principle) be the one that determines the enantioselectivity of the reaction. Adding the energy profile for the other enantiomer will clarify this aspect.

Considering the many approximations (necessarily) included in the calculations, I suggest to replace statements like "Computations demonstrated that the enantioselectivity originates from the significant structural differences between the S- and R-pathways." with something like "Computations suggested that the enantioselectivity originates from the significant structural differences between the S- and R-pathways."

Our response:

We thank the reviewer's comment and suggestion for improving our mechanistic studies. The point-to-point responses are listed as follow:

Suggestion:

"I would suggest to include in the same figure the energy profile for the minor enantiomer as well."

Our response:

We included the computed cycle for the minor enantiomer in the Fig. 5 (right).

Suggestion:

"The authors should also include in the caption the energy unit and a summary of the chosen computational methodology and reaction conditions."

Our response:

The requested changes were made for the caption in the revised manuscript.

Suggestion:

"I suggest to replace statements like "Computations demonstrated that the enantioselectivity originates from the significant structural differences between the S- and R-pathways." with something like "Computations suggested that the enantioselectivity originates from the significant

structural differences between the S- and R-pathways."

Our response:

The sentence was changed according to the reviewer's suggestion.

Additionally, the revised manuscript with tracked changes feature was submitted according to editorial request.

We hope that our revised manuscript satisfies the criteria for publication in *Nature Commun.*

If you have any problems about our revised manuscript, please no hesitate to let me know.

Thank you very much again for your kind help!

Yours sincerely,

Prof. Wanbin Zhang

School of Chemistry and Chemical Engineering

Shanghai Jiao Tong University

800 Dongchuan Road, Shanghai 200240, P. R. China

Phone: +86-21-54743265; Fax: +86-21-54743265

E-mail: wanbin@sjtu.edu.cn

Homepage: wanbin.sjtu.edu.cn